# LEARNING TO PERCEIVE OBJECTS BY PREDICTION

## ABSTRACT

The representation of objects is the building block of higher-level concepts. Infants develop the notion of objects without supervision. The prediction error of future sensory input is likely the major teaching signal for infants. Inspired by this, we propose a new framework to extract object-centric representation from single 2D images by learning to predict future scenes in the presence of moving objects. We treat objects as latent causes whose function to an agent is to facilitate efficient prediction of the coherent motion of their parts in visual input. Distinct from previous object-centric models, our model learns to explicitly infer objects' locations in 3D environment in addition to segmenting objects. Further, the network learns a latent code space where objects with the same geometric shape and texture/color frequently group together. The model requires no supervision or pre-training of any part of the network. We provide a new synthetic dataset with more complex textures on objects and background and found several previous models not based on predictive learning overly rely on clustering colors and lose specificity in object segmentation. Our work demonstrates a new approach for learning symbolic representation grounded in sensation and action.

## 1 INTRODUCTION

Visual scenes are composed of various objects in front of backgrounds. Discovering objects from 2D images and inferring their 3D locations is crucial for planning actions in robotics (Devin et al., 2018; Wang et al., 2019) and this can potentially provide better abstraction of the environment for reinforcement learning (RL), e.g. Veerapaneni et al. (2020). The appearance and spatial arrangement of objects, together with the lighting and the viewing angle, determine the 2D images formed on the retina or a camera. Therefore, objects are latent causes of 2D images, and discovering object is a process of inferring latent causes (Kersten et al., 2004). The predominant approach in computer vision for identifying and localizing objects rely on supervised learning to infer bounding boxes (Ren et al., 2015; Redmon et al., 2016) or pixel-level segmentation of objects (Chen et al., 2017). However, the supervised approach requires expensive human labeling. It is also difficult to label every possible category of objects. Therefore, an increasing interest has developed recently to build unsupervised or self-supervised models to infer objects from images, such as MONet (Burgess et al., 2019), IODINE (Greff et al., 2019) slot-attention (Locatello et al., 2020), GENESIS (Engelcke et al., 2019; 2021), C-SWM (Kipf et al., 2019) and mulMON (Nanbo et al., 2020). Our work also focuses on the same unsupervised *object-centric representation learning* (OCRL) problem, but offers a new learning objective and architecture to overcome the limitation of existing works in segmenting more complex scenes and explicitly represents objects' 3D locations.

The majority of the existing OCRL works are demonstrated on relatively simple scenes with objects of pure colors and background lacking complex textures. As recently pointed out, the success of several recent models based on a variational auto-encoder (VAE) architecture (Kingma & Welling, 2013; Rezende et al., 2014) depends on a reconstruction bottleneck that needs to be intricately balanced (Engelcke et al., 2020). To evaluate how such models perform on scenes with more complex surface textures, we created a new dataset of indoor scenes with diverse texture patterns on the objects and background. We found that several existing unsupervised OCRL models overly rely on clustering pixels based on their colors. A challenge in our dataset and in real-world perception is that sharp boundaries between different colors exist both at contours of objects and within the surface of the same object. A model essentially has to implicitly learn a prior knowledge of which types of boundaries are more likely to be real object contours. The reconstruction loss in existing works appears to be insufficient for learning this prior.

To tackle this challenge, we draw our inspiration from development psychology and neuroscience. Infants understand the concept of object as early as 8 months, before they can associate objects with names (Piaget & Cook, 1952; Flavell, 1963). The fact that infants are surprised when object are hidden indicates that they have already learned to segment discrete objects from the scene and that their brains constantly make prediction and check for deviation from expected outcome. As the brain lacks direct external supervision for object segmentation, the most likely learning signal is from the error of this prediction. In the brain, a copy of the motor command (efference copy) is sent from the motor cortex simultaneously to the sensory cortex, which is hypothesized to facilitate the prediction of changes in sensory input due to self-generated motion (Feinberg, 1978). What remains to be predicted are changes in visual input due to the motion of external objects. Therefore, we believe that the functional purpose of grouping pixels into object is to allow the prediction of the motion of the constituting pixels in a coherent way by tracking very few parameters (e.g., the location, pose, and speed of an object). Driven by this hypothesis, our contribution in this paper is: (1) we combine predictive learning and explicitly 3D motion prediction to learn 3D aware object-centric representation, which we call Object Perception by Predictive LEarning (OPPLE); (2) we provide a new dataset[1] with complex surface texture and motion by both the camera and objects to evaluate object-centric representation models; (3) we found several previous models overly rely on clustering colors to segment objects; (4) although our model leverages image prediction as learning objective, the architecture generalize the ability of object segmentation and spatial localization to single-frame images.

## 2 METHOD

### 2.1 PROBLEM FORMULATION

We denote a scene as a set of distinct objects and a background $\mathbb{S} = \{O_1, O_2, \ldots, O_K, B\}$, where $K$ is the number of objects in scene. At any moment $t$, we denote two state variables, the location and pose of each object relative to the perspective of an observer (camera), as $\boldsymbol{x}_{1:K}^{(t)}$ and $\boldsymbol{\phi}_{1:K}^{(t)}$, where $\boldsymbol{x}_k^{(t)}$ is the 3-d coordinate of the $k$-th object and $\phi_k^{(t)}$ is its yaw angle from a canonical pose, as viewed from the reference frame of the camera (for simplicity, we do not consider pitch and roll here and leave it for future work to extend to 3D pose). At time $t$, given the location of the camera $\boldsymbol{o}^{(t)} \in \mathbb{R}^3$ and its facing direction $\alpha^{(t)}$, $\mathbb{S}$ renders a 2D image on the camera as $\boldsymbol{I}^{(t)} \in \mathbb{R}^{w \times h \times 3}$, where $w \times h$ is the size of the image. Our goal is to train a neural network that infers properties of objects given only a single image $\boldsymbol{I}^{(t)}$ as the sole input without external supervision and with only the information of the intrinsics and egomotion of the camera without explicit knowledge of its extrinsic matrix:

$$\{\boldsymbol{z}_{1:K}^{(t)}, \boldsymbol{\pi}_{1:K+1}^{(t)}, \hat{\boldsymbol{x}}_{1:K}^{(t)}, \boldsymbol{p}_{\phi_{1:K}}^{(t)}\} = f_{\text{obj}}(\boldsymbol{I}^{(t)}) \tag{1}$$

Here, $\boldsymbol{z}_{1:K}^{(t)}$ is a set of view-invariant vectors representing the identity of each object $k$. "View-invariant" is loosely defined as $|\boldsymbol{z}_k^{(t)} - \boldsymbol{z}_k^{(t+\Delta t)}| < |\boldsymbol{z}_k^{(t)} - \boldsymbol{z}_l^{(t)}|$ for $k \neq l$ and $\Delta t > 0$ in most cases, i.e., the vector codes are more similar for the same object across views than they are different across objects. $\boldsymbol{\pi}_{1:K+1}^{(t)} \in \mathbb{R}^{(K+1) \times w \times h}$ are the probabilities that each pixel belongs to any of the objects or the background ($\sum_k \pi_{kij} = 1$ for any pixel at $i, j$), which achieve object segmentation. To localize objects, $\hat{\boldsymbol{x}}_{1:K}^{(t)}$ are the estimated locations of each object relative to the observer and $\boldsymbol{p}_{\phi_{1:K}}^{(t)}$ are estimated probability distribution of the poses of each object. Each $\boldsymbol{p}_{\phi_k}^{(t)} \in \mathbb{R}^b$ is a probability distribution over $b$ equally-spaced bins of yaw angles in $(0, 2\pi)$.

### 2.2 OVERALL PRINCIPLE: LEARNING OBJECT REPRESENTATION BY PREDICTING THE FUTURE

Our hypothesis is that the notion of object emerges to meet the need of efficiently predicting the future fates of all parts of an object. With the (to be learned) ability to infer an object's pose and location from each frame, the object's speed of translation and rotation can be estimated from consecutive frames. If depth is further inferred for each pixel belonging to an object, then the optical

---

[1]We will release upon publication of the paper

flow of each pixel can be predicted based on the object's speed and the position of each pixel relative to the object's center. The pixel-segmentation of an object essentially prescribes which pixels should move together with the object. With the predicted optical flow, one can further predict part of the next image by warping the current image. The parts of the next image unpredictable by warping include surfaces of objects or the background that are currently occluded but will become visible, and the region of a scene newly entering the view due to self- or object-motion. These portions can only be predicted based on the learned statistics of the appearance of objects and background, which we call "imagination". In this work, we will show that with the information of self-motion, knowledge of geometry (rule of rigid-body movement) and the assumption of smooth object movement, the object representations captured by function $f_{obj}$ and depth perception can be learned without supervision.

## 2.3 NETWORK ARCHITECTURE

To demonstrate the hypothesized principle above, we build our OPPLE networks as illustrated in Figure 1, which process two consecutive images individually and make prediction for the next image with the information extracted from them.

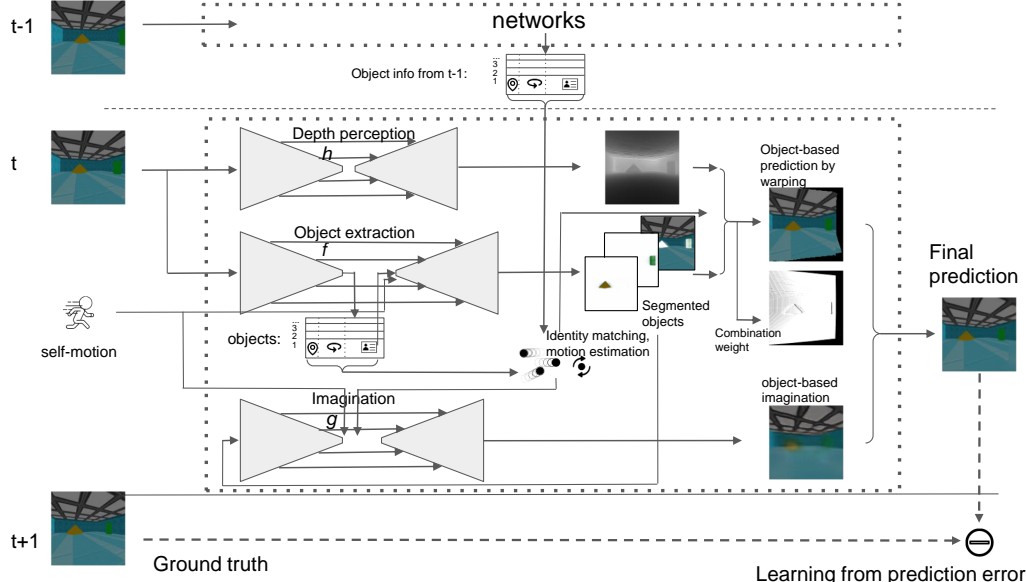

Figure 1: **Architecture for the Object Perception by Predictive LEarning (OPPLE) network** Images at each time point are processed by the Object Extraction and Depth Perception Networks independently. All networks use U-Net structure. The dotted boxes indicate the entire OPPLE network acting at each time step (details omitted for $t-1$). Motion information of each object is estimated from the spatial information extracted for each object between $t-1$ and $t$. Objects between frames are soft-matched by a score depending on the distance between their latent codes. Self- and object motion information are used together with object segmentation and depth map to predict the next image by warping the current image. The segmented object images and depth, together with their motion information and the observer's motion, are used by the imagination network to imagine the next scene and fill the gap not predictable by warping. The error between the final combined prediction and the ground truth of the next image provides teaching signals for all three networks.

**Object extraction network.** We build an object extraction network $f_{\theta_{obj}}$ by modificaiton of a U-Net (Ronneberger et al., 2015) to extract representation for each object, the pixels it occupies and its spatial location and pose. A basic U-Net is composed of a convolutional encoder and a transposed convolutional decoder, while each encoder layer sends a skip connection to the corresponding decoder layer, so that the decoder can combine both global and local information. Inside our $f_{\theta_{obj}}$, an image $\boldsymbol{I}^{(t+1)}$ first passes through the encoder. Additional Atrous spatial pyramid pooling layer (Chen et al., 2017) is inserted between the middle two convolutional layers of the encoder to expand the receptive field. The top layer of the encoder outputs a feature vector $e^t$ capturing the global information of the scene. A Long-Short Term Memory (LSTM) network further repeatedly reads in $e^{(t)}$

and sequentially outputs one code for an object at a time. Each object code is then mapped through a one-layer fully connected network to predict object code $\boldsymbol{z}_k^{(t)}$, object location $\hat{\boldsymbol{x}}_k^{(t)}$ and object pose probability $\boldsymbol{p}_{\phi_k^{(t)}}$, $k = 1, 2, \cdots, K$. The inferred location is restricted to the visible range of the camera with an upper limit of distance. The pose prediction is represented as $\log \boldsymbol{p}_{\phi_k^{(t)}}$ for numerical stability. Each object code vector $\boldsymbol{z}_k^{(t)}$ is then independently fed through the decoder with shared skip connection from the encoder. The decoder outputs one channel for each pixel, representing an un-normalized log likelihood that the pixel belongs to the object $k$. The unnormalized logit maps for all objects are concatenated with a map of all zero for the background, and compete through a softmax function to output the probabilistic segmentation map $\boldsymbol{\pi}_k^{(t)}$.

**Depth perception network.** We use a standard U-Net for depth perception function $h_{\theta_{\text{depth}}}$ that processes images $\boldsymbol{I}^{(t)}$ and output a single-channel depth map $\boldsymbol{D}^{(t)}$.

**Object-based imagination network.** We build the imagination network $g_{\theta_{\text{imag}}}$ also with a modified U-Net. The input is concatenated image $\boldsymbol{I}^{(t)}$ and log of depth $log(\boldsymbol{D}^{(t)})$ inferred by the depth perception network, both multiplied element-wise by one probabilistic mask $\boldsymbol{\pi}_k^{(t)}$. The output of the encoder network is concatenated with a vector composed of the observer's moving velocity $\boldsymbol{v}_{\text{obs}}^{(t)}$ and rotational speed $\boldsymbol{\omega}_{\text{obs}}^{(t)}$, and the estimated object location $\hat{\boldsymbol{x}}_k^{(t)}$, velocity $\hat{\boldsymbol{v}}_k^{(t)}$ and rotational speed $\hat{\boldsymbol{\omega}}_k^{(t)}$ before entering the decoder. The decoder outputs five channels for each pixel: three for predicting RGB colors, one for depth and one for the probability of the pixel belonging to any object $k$ or background and is used to weight the predicted color and depth for the final "imagination".

## 2.4 Learning object representation by prediction

Below, we explain details of the approach. The pseudocode of the algorithm is provided in the appendix.

### 2.4.1 Prediction by warping

We first describe the prediction of part of the next image by warping the current image. Here we consider only rigid objects and the fates of all visible pixels belonging to an object. With depth $\boldsymbol{D}^{(t)} \in \mathbb{R}^{w \times h} = h_\theta(\boldsymbol{I}^{(t)})$ of all pixels in a view inferred by the Depth Perception network based on visual features in the image $\boldsymbol{I}^{(t)}$, the 3D location of a pixel $\hat{\boldsymbol{m}}_{(i,j)}^{(t)}$ at any coordinate $(i, j)$ in the image, can be determined given the focal length $d$ of the camera. We use the inferred current and previous locations, $\hat{\boldsymbol{x}}_k^{(t)}$ and $\hat{\boldsymbol{x}}_k^{(t-1)}$ to estimate the instantaneous velocity of the $K^{th}$ object $\hat{\boldsymbol{v}}_k^{(t)} = \hat{\boldsymbol{x}}_k^{(t)} - \hat{\boldsymbol{x}}_k^{(t-1)}$ for all pixels $(i, j)$ attributed to that $K^{th}$ object. Similarly, with the inferred the current and previous pose probabilities of the object, $\boldsymbol{p}_{\phi_k}^{(t)}$ and $\boldsymbol{p}_{\phi_k}^{(t-1)}$, we can obtain the likelihood of its angular velocity $p(\phi_k^{(t)}, \phi_k^{(t-1)} \mid \omega_k^{(t)} = \omega)$. We obtain the posterior distribution of the object's next pose $p(\phi_k^{(t+1)} \mid \phi_k^{(t)}, \phi_k^{(t-1)})$ with a Von Mises prior. More details about warping and pose estimation are added in the Appendix.

Assuming a pixel $(i, j)$ belongs to object $k$, we use the estimated motion information $\hat{\boldsymbol{v}}_k^{(t)}$ and $p(\omega_k^{(t)} \mid \phi_k^{(t)}, \phi_k^{(t-1)})$ of the object, together with the current location and pose of the object and the current 3D location $\hat{\boldsymbol{m}}_{(i,j)}^{(t)}$ of the pixel, to predict the 3D location $\boldsymbol{m}'_{k,(i,j)}^{(t+1)}$ of the pixel at the next moment as $\boldsymbol{m}'_{k,(i,j)}^{(t+1)} = \boldsymbol{M}_{-\omega_{\text{obs}}}^{(t)}[\boldsymbol{M}_{\hat{\omega}_k}^{(t)}(\hat{\boldsymbol{m}}_{(i,j)}^{(t)} - \hat{\boldsymbol{x}}_k^{(t)}) + \hat{\boldsymbol{x}}_k^{(t)} + \hat{\boldsymbol{v}}_k^{(t)} - \boldsymbol{v}_{\text{obs}}^{(t)}]$, where $\boldsymbol{M}_{-\omega_{\text{obs}}}^{(t)}$ and $\boldsymbol{M}_{\hat{\omega}_k}^{(t)}$ are rotational matrices due to the rotation of the observer and the object, respectively, and $\boldsymbol{v}_{\text{obs}}^{(t)}$ is the velocity of the observer (relative to its own reference frame at $t$). In this way, assuming objects move smoothly most of the time, if the self motion information is known, the 3D location of each visible pixel can be predicted. If a pixel belongs to the background, $\omega_{K+1} = 0$ and $\boldsymbol{v}_{K+1} = 0$ ($K + 1$ is the background's index). Given the predicted 3D location, the target coordinate $(i', j')_k^{(t+1)}$ of the pixel on the image and its new depth $D'_k(i, j)^{(t+1)}$ can be calculated. This prediction of pixel movement allows predicting the image $\boldsymbol{I}'^{(t+1)}$ and depth $\boldsymbol{D}'^{(t+1)}$ by weighting the colors and depth of pixels predicted to land near each pixel at the discrete grid of the next frame.

Considering that object attribution of each pixel has to be inferred, we can write object attribution as a probability of belonging to each object, across all pixels, $\boldsymbol{\pi}_k^{(t)}$, $k = 1, 2, \cdots, K+1$. The predicted motion of each pixel should be described as a probability distribution over $K+1$ discrete target locations $p(\boldsymbol{x}'^{(t+1)}_{(i,j)}) = \sum_{k=1}^{K+1} \pi_{kij}^{(t)} \cdot \delta(\boldsymbol{x}'^{(t+1)}_{k,(i,j)})$, i.e., pixel $(i, j)$ has a probability of $\pi_{kij}^{(t)}$ to move to location $\boldsymbol{x}'^{(t+1)}_{k,(i,j)}$. Using probabilistic prediction of pixel movements, we partially predict the color of next image at the pixel grids where some original pixels from the current will land by weighting their contribution $\boldsymbol{I}'^{(t+1)}_{\text{Warp}}(p, q)$. This contribution term is used to calculate a weight for effect of a pixel (i,j) on position (p,q) on the grid, and is written as $w_k(i, j, p, q)$. More details in the appendix.

### 2.4.2 IMAGINATION

For the regions not fully predictable by warping current image with appendix equation (7), i.e., for $(p, q)$ where $\sum_{k,i,j} w_k(i, j, p, q) < 1$, we learn a function $g$ that "imagines" the appearance $\boldsymbol{I}'^{(t+1)}_{k\text{Imag}} \in \mathbb{R}^{w \times h \times 3}$ and the pixel-wise depth $\boldsymbol{D}'^{(t+1)}_{k\text{Imag}} \in \mathbb{R}^{w \times h}$ of the object or background $k$ in the next frame, and the predicted probabilities that each pixel in the next frame belongs to each object or the background $\boldsymbol{\pi}'^{(t+1)}_{k\text{Imag}} \in \mathbb{R}^{w \times h}$. The function takes as input portion of the current image corresponding to each object $\boldsymbol{I}^{(t)} \odot \boldsymbol{\pi}_k^{(t)}$ and the inferred depth $\boldsymbol{D}^{(t)} \odot \boldsymbol{\pi}_k^{(t)}$, both extracted by element-wise multiplying with the probabilistic segmentation mask $\boldsymbol{\pi}_k^{(t)}$, the information of the camera's self motion, and the location and motion of that object:

$$\{\boldsymbol{I}'^{(t+1)}_{k\text{Imag}}, \boldsymbol{D}'^{(t+1)}_{k\text{Imag}}, \boldsymbol{\pi}'^{(t+1)}_{k\text{Imag}}\} = g(\boldsymbol{I}_i^{(t)} \odot \boldsymbol{\pi}_k^{(t)}, \boldsymbol{D}^{(t)} \odot \boldsymbol{\pi}_k^{(t)}, \hat{\boldsymbol{x}}_k^{(t)}, \hat{\boldsymbol{v}}_k^{(t)}, \hat{\omega}_k^{(t)}, \omega_{\text{obs}}^{(t)}) \tag{2}$$

The "imagination" specific for each object and the background can then be merged using the weights prescribed by $\boldsymbol{\pi}'^{(t+1)}_{1:K\text{Imag}}$: $\boldsymbol{I}'^{(t+1)}_{\text{Imag}} = \sum_k \boldsymbol{I}'^{(t+1)}_{k\text{Imag}} \odot \boldsymbol{\pi}'^{(t+1)}_{k\text{Imag}}$, and $\boldsymbol{D}'^{(t+1)}_{\text{Imag}} = \sum_k \boldsymbol{D}'^{(t+1)}_{k\text{Imag}} \odot \boldsymbol{\pi}'^{(t+1)}_{k\text{Imag}}$.

### 2.4.3 COMBINING WARPING AND IMAGINATION

The final predicted image or depth map are weighted average of the prediction made by warping the current image or predicted depth map and the corresponding predictions by imagination:

$$\boldsymbol{I}'^{(t+1)} = \boldsymbol{I}'^{(t+1)}_{\text{Warp}} \odot \boldsymbol{W}_{\text{Warp}} + \boldsymbol{I}'^{(t+1)}_{\text{Imag}} \odot (1 - \boldsymbol{W}_{\text{Warp}}) \tag{3}$$

Here, $\boldsymbol{W}_{\text{Warp}} \in \mathbb{R}^{w \times h}$ with each element $W_{\text{Warp}}(p, q) = \max\{\sum_{k,i,j} w(i, j, p, q), 1\}$. The intuition is that imagination is only needed when there is not sufficient contribution for predicting a pixel by warping. The same weighting applies for generating the final predicted depth $\boldsymbol{D}'^{(t+1)}$.

In addition to the image, the states of the objects can also be predicted. The location of object $k$ at $t + 1$ can be predicted as $\boldsymbol{x}'^{(t+1)}_k = \boldsymbol{M}^{(t)}_{-\omega_{\text{obs}}}(\hat{\boldsymbol{x}}_k^{(t)} + \hat{\boldsymbol{v}}_k^{(t)} - \boldsymbol{v}_{\text{obs}}^{(t)})$. Its new pose probability can be predicted by $p'^{(t+1)}(\phi_k + \omega_{\text{obs}} = \gamma_2) = \sum_{\substack{\gamma_1, \omega, \gamma_2 - \gamma_1 \in \\ \{\omega - 2\pi, \omega, \omega + 2\pi\}}} p(\hat{\omega}_k^{(t)} = \omega) p(\phi_k^{(t)} = \gamma_1)$ for $\gamma_2$ equal to each fixed yaw angle bin. To obtain $p'^{(t+1)}(\phi_k)$ instead of $p'^{(t+1)}(\phi_k + \omega_{\text{obs}})$ at the same set of bins, the vector $p'^{(t+1)}(\phi_k + \omega_{\text{obs}})$ can be shifted by multiplication with a pre-computed matrix composed of a bank of shifted Von Mises distributions to "move" the probability mass on the angle bins.

There is one important issue of object-centric representation when making prediction for future images: in order to predict the spatial state of each object at $t + 1$ based on the views at $t$ and $t - 1$, the network needs to match the representation of an object at $t$ from the representation of the same object at $t - 1$. As the dimensions of features (e.g., shape, surface texture, size, etc.) grows, the number of possible objects grows exponentially. Therefore, we cannot simply match object representations based on the order by which an LSTM extracts objects, as this requires learning a consistent order over enormous amount of objects. Instead, we take a soft-matching approach: we take a subset of the features in $\boldsymbol{z}_k^{(t)}$ extracted by $f$ as an identity code for each object. For object $k$ at time $t$, we calculate the distance between its identity code and those of all objects at $t - 1$, and pass the distances through a radial basis function to serve as a matching score $r_{kl}$ indicating how closely the object $k$ matches each of the previous objects. The scores are used weight all the

estimated translational and rotational speeds for object $k$ each assuming a different object $l$ were the true object $k$ at $t-1$. We additionally introduce a fixed identity code $\boldsymbol{z}_{K+1} = 0$ for the background and set the predicted motion of background to zero.

### 2.4.4 LEARNING OBJECTIVE

Above, we have explained how the next image input $\boldsymbol{I}'^{(t+1)}$, the depth map $\boldsymbol{D}'^{(t+1)}$ and the spatial states of each object, $\boldsymbol{x}'^{(t+1)}_k$ and $p'^{(t+1)}_{\phi_k}$ can be predicted based on object-centric representation extracted by a function $f$ from the current and previous images $\boldsymbol{I}^{(t)}_i$ and $\boldsymbol{I}^{(t-1)}_i$, the depth $\boldsymbol{D}'^{(t)}$ extracted by a function $h$, combined with the prediction from object-based imagination function $g$ that are all to be learned. Among the three prediction targets, only the ground truth of visual input $\boldsymbol{I}^{(t+1)}$ is available, while the other can only be inferred by $f$ and $h$ from $\boldsymbol{I}^{(t+1)}$. Therefore, for the prediction targets other than $\boldsymbol{I}^{(t+1)}$, we use the self-consistent loss between the predicted value based on $t$ and $t-1$ and the inferred value based on $t+1$ as additional regularization terms to learn the functions $f$ and $g$.

To learn the functions $f$, $g$ and $h$, we approximate them with deep neural networks with parameters $\theta$ and optimize $\theta$ to minimize the following loss function:

$$L = L_{\text{image}} + \lambda_{\text{depth}} L_{\text{depth}} + \lambda_{\text{spatial}} L_{\text{spatial}} + \lambda_{\text{map}} L_{\text{map}} \tag{4}$$

Here, $L_{\text{image}} = \text{MSE}(I'^{(t+1)}, I^{(t+1)})$ is the image prediction error. $L_{\text{depth}} = \text{MSE}(\log(D'^{(t+1)}), \log(\hat{D}^{(t+1)}))$ is the error between the predicted and inferred depth. These provide major teaching signals. $L_{\text{spatial}} = \sum_{k=1}^{K} |\sum_{l=1}^{K+1} r_{kl} \boldsymbol{x}'^{(t+1)}_l - \hat{\boldsymbol{x}}^{(t+1)}_k|^2 - \sum_{k=1}^{K} \min\{|\hat{\boldsymbol{x}}^{(t+1)}_{\text{rand}} - \hat{\boldsymbol{x}}^{(t+1)}_k|, \delta\} + \sum_{k=1}^{K} |\hat{\boldsymbol{x}}^{(t+1)}_k - \sum_{i,j} \hat{\boldsymbol{m}}^{(t+1)}_{i,j} \pi_{kij}|^2 + \sum_{k=1}^{K} D_{\text{KL}}(\hat{\boldsymbol{p}}^{(t+1)}_{\phi_k} || \sum_{l=1}^{K+1} r_{kl} \boldsymbol{p}'^{(t+1)}_{\phi_l})$ is the self-consistent loss on spatial information prediction. The first term is the error between inferred and predicted location of each object, while the calculation of the predicted location incorporates soft matching between objects in consecutive frames. The second term is the negative term of contrastive loss, which we found empirically prevents the network from reaching a local minimum where all objects are inferred at the same location relative to the camera (and covering minimal regions of the picture). $\hat{\boldsymbol{x}}_{\text{rand}}$ is the inferred object location from a random sample within the same batch. The third term penalizes the discrepancy between the inferred object location and the average location of pixels in its segmentation mask. The last term is the KL-divergence between the predicted and inferred pose for each object at $t+1$. $L_{\text{map}} = \text{ReLu}(10^{-4} - \boldsymbol{\pi}_{1:K}) + \boldsymbol{\pi}_k \cdot \boldsymbol{\pi}_l$, for $k \neq l$ avoids loss of gradient due to zero probability of object belonging and discourages overlap between maps of different objects.

### 2.5 DATASET

We procedurally generated a dataset composed of 306445 triplets of images captured by a virtual camera with field of view of 90 degrees in a square room. The camera translates horizontally and pans with random small steps between consecutive frames to facilitate the learning of depth perception. 3 objects with random shape, size, surface color or textures are spawned at random locations in the room and each move with a randomly selected constant velocity and panning speed. The translation and panning of the the camera is known to the networks. No other ground truth information is provided. The first two frames serve as data and the last frame serve as the prediction target at $t+1$. An important difference between this dataset and other commonly used synthetic datasets for OCRL is that more complex and diverse textures are used on both the objects and the background. We further evaluated on a richer version of the Traffic dataset (Henderson & Lampert, 2020) in A.6.

### 2.6 COMPARISON WITH OTHER WORKS

To compare our work with the states-of-the-art models of unsupervised object-centric representation learning, we trained MONet (Burgess et al., 2019), slot-attention (Locatello et al., 2020) and GENESIS[2] (Engelcke et al., 2021) on the same dataset. Although these models are trained on single images, all images of each triplets are used for training.

---

[2]We failed to obtain reasonable result by training GENESIS V2 on our dataset, thus we adopted a GENESIS network pre-trained on GQN dataset and retrained on our dataset with K=7.

To address the concern that the original configurations of the models are not optimized for more difficult dataset, we trained variants of some of the models with large network size. For MONet, we tested channel numbers of [32, 64, 128, 128] (MONet-128) and [32, 64, 128, 256, 256] (MONet-128-bigger) for the hidden layers of encoder of the component VAE instead of [32, 32, 64, 64] and adjusted decoder layers sizes accordingly, and increased the base channel from 64 to 128 for the attention network. For slot attention, we tested a variant which increased the number of features in the attention component from 64 to 128 (slot-attention-128). Slot numbers were chosen as 4 except for GENESIS.

| Model | ARI-fg | IoU |
|---|---|---|
| MONet | 0.31 | 0.08 |
| MONet-128 | 0.33 | 0.22 |
| MONet-128-bigger | 0.33 | 0.15 |
| slot-attention | 0.41 | 0.31 |
| slot-attention-128 | 0.39 | **0.54** |
| GENESIS | 0.17 | 0.03 |
| our model (OPPLE) | **0.46** | 0.35 |

Table 1: Performance of different models on object segmentation.

## 3 RESULTS

After training the networks, we evaluate them on 4000 test images unused during training but generated randomly with the same procedure, thus coming from the same distribution. We compare the performance of different models mainly on their segmentation performance. Additionally, we demonstrate the ability unique to our model: inferring locations of objects in 3D space and the depth of the scene. The performance of depth perception is illustrated in the appendix.

### 3.1 OBJECT SEGMENTATION

Following prior works (Greff et al., 2019; Engelcke et al., 2019; 2021), we evaluated segmentation with the Adjusted Rand Index of foreground objects (ARI). In addition, for each image, we matched ground-true objects and background with each of the segmented class by ranking their Intersection over Union (IoU) and quantified the average IoU over all foreground objects[3]. The performance is summarized in table 2.6 .

Our model outperforms all compared models on ARI and is second to a slot-attention-128 in IoU. As shown in Figure 2, MONet and GENESIS appear to heavily rely on color to group pixels into the same masks. Even though some of these models almost fully designate pixels of an object to a mask, the masks lacks specificity in that they often include pixels with similar colors from other objects or background. Patterns on the backgrounds are often treated as objects as well. The reason of such drawbacks awaits further investigation but we postu-

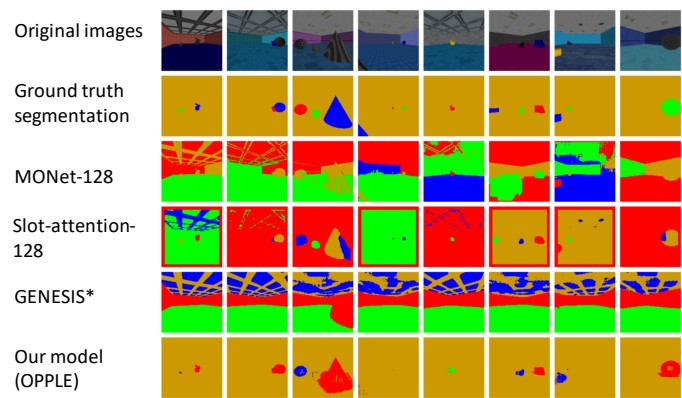

Figure 2: Example of object segmentation by different models

late there may be fundamental limitation in the approach that learns purely from static discrete images. Patches in the background with coherent color offer room to compress information similarly as objects with coherent colors do, and their shapes re-occur across images just as other objects. Our model is able to learn object-specific masks because these masks are used to predict optical flow specific to each object. A wrong segmentation would generate large prediction error even if the motion of an object is estimated correctly. Such prediction error forces the masks to be concentrated on object surface. They emerge first at object boundaries where the prediction error is the largest and gradually grow inwards during training. Figure 3A-D further compares the distribution

---

[3]Due to the artifact at the border of slot-attention model, IoU was calculated excluding those border pixels

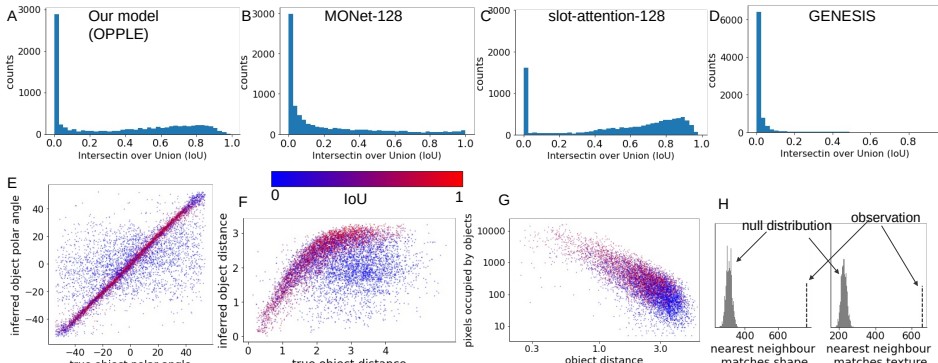

Figure 3: **A-D**: distribution of IoU. All models have IoU $< 0.01$ for about 1/4 of objects. Only OPPLE shows a bi-modal distribution while other models' IoU are more skewed towards 0. **E-F**: object localization accuracy of OPPLE for object's polar angle and distance relative to the camera. Each dot is a valid object with color representing its segmentation IoU. Angle estimation is highly accurate for well segmented objects (red dots). Distance is under-estimated for farther objects. **G**: objects with failed segmentation (blue dots) are mostly far away and occupying few pixels. **H** The numbers of objects sharing the same shape or texture with their nearest neighbour objects in latent space are significantly above chance.

of IoU across models. Most models have IoU $< 0.01$ for about 1/4 of objects. Only OPPLE and slot-attention-128 show bi-modal distributions while other models' IoU are more skewed towards 0. Figure 3G plots each object's distance and size on the picture with colors corresponding to their IoUs in our model. Objects with poor segmentation (blue dots) are mostly far away from the camera and occupy few pixels. This is reasonably because motion of farther objects causes less shift on the images and thus provide weaker teaching signal for the network to assign their pixels as separate from the background. For other models, blue dots are more spread even for near objects (not shown).

## 3.2 OBJECT LOCALIZATION

The Object Extraction Network infers object location relative to the camera. We convert the inferred locations to angles and distance in polar coordinate relative to the camera. Figure 3E-F plot the true and inferred angles and distance, color coded by objects' IoUs. For objects well segmented (red dots), their angles are estimated high accurately (concentrated on the diagonal in E). Distance estimation is negatively biased for farther objects, potentially because the regularization term on the distance between the predicted and inferred object location at frame $t + 1$ favors shorter distance when estimation is noisy. Note that the ability to explicitly infer object's location is not available in other models compared.

## 3.3 MEANINGFUL LATENT CODE

Because a subset of the latent code (10 dimensions) was used to calculate object matching scores between frames in order to soft-match objects, this should force the object embedding $z$ to be similar for the same objects. We explored the geometry of the latent code by examining whether the nearest neighbours of each of the object in the test data with IoU $> 0.5$ are more likely to have the same property as themselves. 772 out of 3244 objects' nearest neighbour had the same shape (out of 11 shapes) and 660 objects' nearest neighbour had the same color or texture (out of 15). These numbers are 28 to 29 times the standard deviation away from the means of the distribution expected if the nearest neighbour were random (Figure 3H). This suggests the latent code reflects meaningful features of objects. However, texture and shape are not the only factors determining latent code, as we found the variance of code of all objects with the same shape and texture to still be big.

## 4 RELATED WORK

Our work is on the same tracks as two recent trends in machine learning: object-centric representation (Locatello et al., 2020) and self-supervised learning (Chen et al., 2020). We take the same

logic as self-supervised learning that learning to predict part of the data based on another part forces a neural network to learn important structures in the data. However, most of the existing works in self-supervised learning do not focus on object-based representation, but instead encode the entire scene as one vector. Other works on object-centric representations overcome this by assigning one representation to each object, as we do. Although works such as MONet (Burgess et al., 2019), IO-DINE (Greff et al., 2019), slot-attention (Locatello et al., 2020), GENESIS (Engelcke et al., 2019) and PSGNet (Bear et al., 2020) can also segment objects and some of them can "imagine" complete objects based on codes extracted from occluded objects or draw objects in the correct order consistent with occlusion, few works explicitly infer an object's location in 3D space together with segmentation purely by self-supervised learning, with the exception of a closely related work O3V (Henderson & Lampert, 2020). Both our works learn from videos. One major distinction is that O3V interleaves spatial and temporal convolution, thus it still require video as input at test time. In contrast, our three major networks process each image independently. Therefore, once trained, our network can generalize to single images. Another distinction from Henderson & Lampert (2020) and many other works is that our model learns from prediction instead of reconstruction. Contrastive-learning of structured world model (Kipf et al., 2019) also learns object masks and predict their future states by linking each object mask with a node in a Graphic Neural Network (GNN). The order of mapping object slot to nodes of GNN is fixed through time, and the actions to objects are coded with specific associated nodes. This arrangement may become infeasible with combinatorial number of different possible objects as the order of assigning different objects to a limited number of nodes may not be consistent across scenes. We solve this by a soft matching of object representation between different time points, which does not require the RNN in the Object Extraction Network to learn a fixed order of extracting different types of objects. On the neuroscience side, our work is highly motivated by recent works on predictive learning (O'Reilly et al., 2021) which also yields view-invariance representation while self-motion signal is available. O'Reilly et al. (2021) used biologically plausible but less efficient learning and applied their model to an easier dataset with objects without background, and did not learn object localization. We should note that explicit spatial localization and depth perception were not pursued in previous works on self-supervised object-centric learning, and the images in our dataset have significantly richer texture information than those demonstrated in previous works (Burgess et al., 2019; Kipf et al., 2019), making the task more challenging. Although view synthesis is not our central goal, the principle illustrated here can be combined with recent advancement in 3D-aware image synthesis (Wiles et al., 2020).

## 5 DISCUSSION

We provide a new approach to learn object-centric representation that includes explicit spatial localization of objects, object segmentation from image, automatic matching the same objects across scene based on a learned latent code and depth perception as a by-product. All of the information extracted by our networks are learned without supervision and no pre-training on other tasks is involved. The only additional information required is that of observer's self-motion, which is available in the brain as efference copy. This demonstrate the possibility of learning rich embodied information of object, one step toward linking neural networks with symbolic representation in general. We expect future works to develop self-supervised learning model for natural categories beyond simple object identity, building on our work.

The work demonstrates that the notion of object can emerge as a necessary common latent cause of the pixels belonging to the object for the purpose of efficiently explaining away the pixels' coherent movement across frames. In our experiment, object spatial location is inferred more easily than object pose (which we have not fully investigated), thus the predicted warping relies more on object translation than rotation. As a limitation, almost all existing object-centric representation works, including ours, focus on rigid bodies and simple environment. Future works need to explore how to learn object representation for deformable objects, objects with more complex shapes and lighting conditions, and more cluttered environment, towards more realistic application. There is important implication for learning 3D-aware object-based representation, because for such representation to be useful eventually for robotics and RL, agents need to understand an object's spatial relation to itself based on vision. Learning object-centric representation with explicit 3D information is an important step toward embodied representation of the environment.

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

# A APPENDIX

## A.1 PSEUDO CODE OF THE OPPLE FRAMEWORK

---

**Algorithm 1** Developing object-centric representation by predicting future scene

---

Init**Initialize:** Network parameters $\theta$

**Input:** images $\boldsymbol{I}^{(t-1)}, \boldsymbol{I}^{(t)} \in \mathbb{R}^{w \times h \times 3}$, self-motion $\boldsymbol{v}_{\text{obs}}^{(t-1)}, \omega_{\text{obs}}^{(t-1)}, \boldsymbol{v}_{\text{obs}}^{(t)}, \omega_{\text{obs}}^{(t)}$

**Output:** prediction $\boldsymbol{I}'^{(t+1)}$, segmentation $\boldsymbol{\pi}_{1:K+1}^{(t-1)}, \boldsymbol{\pi}_{1:K+1}^{(t)}$, objects' codes $\boldsymbol{z}_{1:K}^{(t-1)}, \boldsymbol{z}_{1:K}^{(t)}$, objects'
    locations and poses $\hat{\boldsymbol{x}}_{1:K}^{(t-1)}, \boldsymbol{p}_{\phi_{1:K}}^{(t-1)}, \hat{\boldsymbol{x}}_{1:K}^{(t)}, \boldsymbol{p}_{\phi_{1:K}}^{(t)}$
    **for** $\tau = \{t-1, t\}$ **do**
        scene code $e^{(\tau)} \leftarrow \text{U-NetEncoder}_{f_\theta}(\boldsymbol{I}^{(\tau)})$
        object code $\boldsymbol{z}_{1:K}^{(\tau)}$, location $\hat{\boldsymbol{x}}_{1:K}^{(\tau)}$, pose $\boldsymbol{p}_{\phi_{1:K}}^{(\tau)} \leftarrow \text{LSTM}_{f_\theta}(e^{(\tau)})$
        background code $\boldsymbol{z}_{K+1} = 0$
        depth $\boldsymbol{D}^{(\tau)} \leftarrow h_\theta(\boldsymbol{I}^{(\tau)})$
        segmentation mask $\boldsymbol{\pi}_{1:K+1}^{(\tau)} \leftarrow \text{Softmax}(\text{U-NetDecoder}_{f_\theta}(\boldsymbol{I}^{(\tau)}, \boldsymbol{z}_{1:K}^{(\tau)}), 0)$
    **end for**
    object matching scores $r_{kl} \leftarrow \text{RBF}(\boldsymbol{z}_k^{(t)}, \boldsymbol{z}_l^{(t-1)}), k, l \in 1 : K+1$
    **for** $k \leftarrow 1$ to $K$ **do**
        object motion $\hat{\boldsymbol{v}}_{1:K}, \boldsymbol{\omega}_{1:K} \leftarrow r_{k,l}, \hat{\boldsymbol{x}}_k^{(t)}, \hat{\boldsymbol{x}}_l^{(t-1)}, \boldsymbol{p}_{\phi_k}^{(t)}, \boldsymbol{p}_{\phi_l}^{(t-1)}, l = 1 : K+1$
        onject-specific optical flow$_k \leftarrow \hat{\boldsymbol{v}}_{1:K}, \boldsymbol{\omega}_{1:K}, \boldsymbol{v}_{\text{obs}}, \omega_{\text{obs}}, \boldsymbol{D}^{(t)}$
    **end for**
    $\boldsymbol{I}'^{(t+1)}_{\text{warp}} \leftarrow \text{Warp}(\boldsymbol{I}^{(t)}, \text{optical flow}_{1:K+1})$
    $\boldsymbol{I}'^{(t+1)}_{\text{imagine}} \leftarrow g_\theta(\boldsymbol{I}^{(t)} \odot \boldsymbol{\pi}_{1:K+1}^{(t)}, log(\boldsymbol{D}^{(t)}) \odot \boldsymbol{\pi}_{1:K+1}^{(t)}, \boldsymbol{v}_{\text{obs}}, \omega_{\text{obs}}, \hat{\boldsymbol{v}}_{1:K+1}, \hat{\boldsymbol{x}}_{1:K})$
    final image prediction: $\boldsymbol{I}'^{(t+1)} \leftarrow \boldsymbol{I}'^{(t+1)}_{\text{warp}}, \boldsymbol{I}'^{(t+1)}_{\text{imagine}}$, warping weights
    update parameters: $\theta \leftarrow \theta - \gamma \nabla_\theta [|\boldsymbol{I}'^{(t+1)} - \boldsymbol{I}^{(t+1)}|^2 + \text{regularization loss}]$

---

## A.2 PERFORMANCE ON DEPTH PERCEPTION

We demonstrate a few example images and the inferred depth. Our network can capture the global 3D structure of the scene, although details on object surfaces are still missing. Because background occurs in every training sample, the network appears to bias the depth estimation on objects towards the depth of the walls behind, as is also shown in the scatter plot.

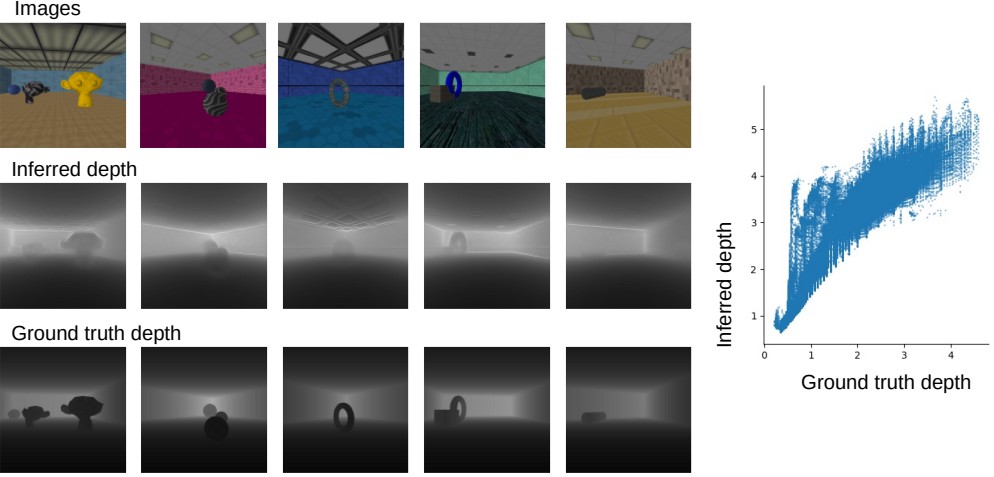

Figure 4: Comparison between ground truth depth and inferred depth

### A.3 NETWORK TRAINING AND DATASET

We trained the three networks jointly using ADAM optimization Kingma & Ba (2014) with a learning rate of $3e-4$, $\epsilon = 1e-6$ and other default setting in PyTorch, with a batch size of 24. 40 epochs were trained on the dataset. We set $\lambda_{\text{spatial}} = 1.0$, $\lambda_{\text{depth}} = 0.1$ and $L_{\text{map}} = 0.005$ Images in datasets are rendered in Unity environment and downsampled to $128 \times 128$ resolution for training. Images are rendered in sequence of 7 steps each time in a room with newly selected texture and object. Camera moves with random steps and random rotation between consecutive frames. All possible triplets equally spaced by 1, 2, and 3 frames form training samples.

The model was implemented in PyTorch and trained on NVidia RTX 6000. We will release the code and dataset upon publication of the manuscript.

### A.4 METHOD

#### A.4.1 PREDICTION BY WARPING

We first describe the prediction of part of the next image by warping the current image. Here we consider only rigid objects and the fates of all visible pixels belonging to an object. With depth $\boldsymbol{D}^{(t)} \in \mathbb{R}^{w \times h} = h_\theta(\boldsymbol{I}^{(t)})$ of all pixels in a view inferred by the Depth Perception network based on visual features in the image $\boldsymbol{I}^{(t)}$, the 3D location of a pixel at any coordinate $(i,j)$ in the image, where $|i| \leq \frac{w-1}{2}, |j| \leq \frac{h-1}{2}$, can be determined given the focal length $d$ of the camera as $\hat{\boldsymbol{m}}^{(t)}_{(i,j)} = \frac{D^{(t)}(i,j)}{\sqrt{i^2+j^2+d^2}} \cdot [i, d, j]$. Here, we take the coordinate of the center of an image as (0,0). On the other hand, with the inferred $\hat{\boldsymbol{x}}^{(t)}_k$ and $\hat{\boldsymbol{x}}^{(t-1)}_k$, the current and previous locations of the object $k$ that the pixel $(i,j)$ belongs to, from $\boldsymbol{I}^{(t)}$ and $\boldsymbol{I}^{(t-1)}$ respectively, we can estimate the instantaneous velocity of the object $\hat{\boldsymbol{v}}^{(t)}_k = \hat{\boldsymbol{x}}^{(t)}_k - \hat{\boldsymbol{x}}^{(t-1)}_k$. Similarly, with the inferred the current and previous pose probabilities of the object, $\boldsymbol{p}^{(t)}_{\phi_k}$ and $\boldsymbol{p}^{(t-1)}_{\phi_k}$, we can obtain the likelihood of its angular velocity

$$p(\phi_k^{(t)}, \phi_k^{(t-1)} \mid \omega_k^{(t)} = \omega) \propto \sum_{\substack{\gamma_1, \gamma_2, \gamma_1 - \gamma_2 \in \\ \{\omega - 2\pi, \omega, \omega + 2\pi\}}} p(\phi_k^{(t)} = \gamma_1) \cdot p(\phi_k^{(t-1)} = \gamma_2) \tag{5}$$

.

By additionally imposing a prior distribution (we use Von Mises distribution) over $\omega_k^{(t)}$ that favors slow rotation, we can obtain the posterior distribution of the object's angular velocity $p(\omega_k^{(t)} \mid \phi_k^{(t)}, \phi_k^{(t-1)})$, and eventually the posterior distribution of the object's next pose $p(\phi_k^{(t+1)} \mid \phi_k^{(t)}, \phi_k^{(t-1)})$.

Assuming a pixel $(i,j)$ belongs to object $k$, using the estimated motion information $\hat{\boldsymbol{v}}^{(t)}_k$ and $p(\omega_k^{(t)} \mid \phi_k^{(t)}, \phi_k^{(t-1)})$ of the object, together with the current location and pose of the object and the current 3D location $\hat{\boldsymbol{m}}^{(t)}_{(i,j)}$ of the pixel, we can predict the 3D location $\boldsymbol{m}'^{(t+1)}_{k,(i,j)}$ of the pixel at the next moment as

$$\boldsymbol{m}'^{(t+1)}_{k,(i,j)} = \boldsymbol{M}^{(t)}_{-\omega_{\text{obs}}} [\boldsymbol{M}^{(t)}_{\hat{\omega}_k} (\hat{\boldsymbol{m}}^{(t)}_{(i,j)} - \hat{\boldsymbol{x}}^{(t)}_k) + \hat{\boldsymbol{x}}^{(t)}_k + \hat{\boldsymbol{v}}^{(t)}_k - \boldsymbol{v}^{(t)}_{\text{obs}}] \tag{6}$$

where $\boldsymbol{M}^{(t)}_{-\omega_{\text{obs}}}$ and $\boldsymbol{M}^{(t)}_{\hat{\omega}_k}$ are rotational matrices due to the rotation of the observer and the object, respectively, and $\boldsymbol{v}^{(t)}_{\text{obs}}$ is the velocity of the observer (relative to its own reference frame at $t$). In this way, assuming objects move smoothly most of the time, if the self motion information is known, the 3D location of each visible pixel can be predicted. If a pixel belongs to the background, $\omega_{K+1} = 0$ and $\boldsymbol{v}_{K+1} = 0$ ($K+1$ is the background's index). Given the predicted 3D location, the target coordinate $(i', j')^{(t+1)}_k$ of the pixel on the image and its new depth $D'_k(i,j)^{(t+1)}$ can be calculated. This prediction of pixel movement allows predicting the image $\boldsymbol{I}'^{(t+1)}$ and depth $\boldsymbol{D}'^{(t+1)}$ by weighting the colors and depth of pixels predicted to land near each pixel at the discrete grid of the next frame, as explained in Sec 2.4.2.

A.4.2 WARPING CONTRIBUTION WEIGHT

As the object attribution of each pixel is not known but is inferred by $f_{\text{obj}}(\boldsymbol{I}^{(t)})$, it is represented for every pixel as a probability of belonging to each object and the background $\boldsymbol{\pi}_k^{(t)}$, $k = 1, 2, \cdots, K + 1$. Therefore, the predicted motion of each pixel should be described as a probability distribution over $K + 1$ discrete target locations $p(\boldsymbol{x}'^{(t+1)}_{(i,j)}) = \sum_{k=1}^{K+1} \pi_{kij}^{(t)} \cdot \delta(\boldsymbol{x}'^{(t+1)}_{k,(i,j)})$, i.e., pixel $(i,j)$ has a probability of $\pi_{kij}^{(t)}$ to move to location $\boldsymbol{x}'^{(t+1)}_{k,(i,j)}$ at the next time point, for $k = 1, 2, \cdots, K+1$. With such probabilistic prediction of pixel movement for all visible pixel $(i,j)^{(t)}$, we can partially predict the colors of the next image at the pixel grids where some original pixels from the current view will land nearby by weighting their contribution:

$$\boldsymbol{I}'^{(t+1)}_{\text{Warp}}(p,q) = \begin{cases} \frac{\sum_{k,i,j} w_k(i,j,p,q) I^{(t)}(i,j)}{\sum_{k,i,j} w_k(i,j,p,q)}, & \text{if } \sum_{k,i,j} w_k(i,j,p,q) > 0 \\ 0, & \text{otherwise} \end{cases} \tag{7}$$

We define the weight of the contribution from any source pixel $(i,j)$ to a target pixel $(p,q)$ as

$$w_k(i,j,p,q) = \pi_{kij}^{(t)} \cdot e^{-\beta \cdot D'^{(t+1)}_k(i,j)} \cdot \max\{1 - |i'^{(t+1)}_k - p|, 0\} \cdot \max\{1 - |j'^{(t+1)}_k - q|, 0\} \tag{8}$$

The first term incorporates the uncertainty of which object a pixel belongs to. The second term $e^{-\beta \cdot D'^{(t+1)}_k(i,j)}$ resolves the issue of occlusion when multiple pixels are predicted to move close to the same pixel grid by down-weighting the pixels predicted to land farther from the camera. These last two terms mean that only the source pixels predicted to land within a square of of $2 \times 2$ pixels centered at any target location $(p,q)$ will contribute to the color $I'^{(t+1)}_{\text{Warp}}(p,q)$. The depth map $\boldsymbol{D}'^{(t+1)}_{\text{Warp}}$ can be predicted by the same weighting scheme after replacing $I^{(t)}(i,j)$ with each predicted depth $D'^{(t+1)}_k(i,j)$ assuming the pixel belongs to object $k$.

A.5 DEPENDENCY OF SEGMENTATION PERFORMANCE ON OBJECT SIZE AND DISTANCE ACROSS MODELS

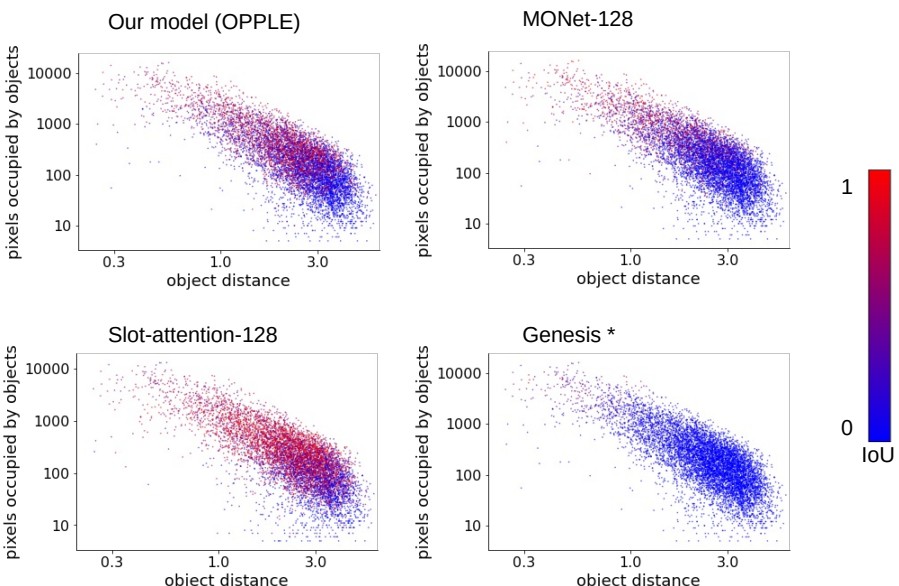

A.6 ILLUSTRATION OF PREDICTION QUALITY

In the figure below, we display the first, second image, one of the masked objects, the predicted third image, and the ground truth of the third image, both for our dataset, and for a richer version of the Traffic dataset used by (Henderson & Lampert, 2020). We provide reference lines and some circles to aid the comparison between images and evaluate the warping quality. Imagination quality can be inspected usually at one side of the predicted images (the camera motion is typically larger in our dataset).

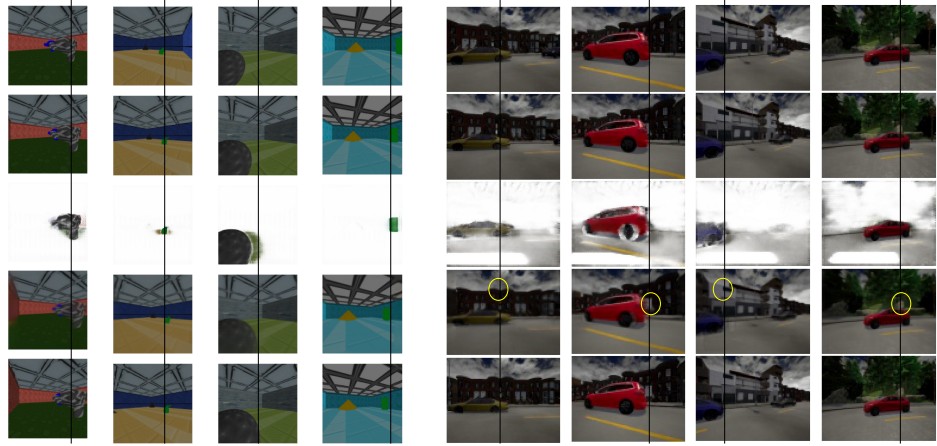

