# OpenReview forum: "Learning to perceive objects by prediction"
_ICLR.cc/2022/Conference — ICLR 2022 Submitted_

### Official Review · Reviewer_uocA · 2021-11-02

**Correctness:** 3
**Technical Novelty And Significance:** 3
**Empirical Novelty And Significance:** 3
**Recommendation:** 8
**Confidence:** 3

**Main Review:**

Strengths

I like the general thrust of this paper. While the basic idea has been around in cognitive psychology, which inspired the authors, I am not aware of any significant implementation of it. This is a nice first step.

To make this work, the authors develop some algorithmic bits that might be useful for followup work.

The results compare well to others in this space, showing that the training strategy has real promise. In some sense, the methods are not really comparable as the other methods do not have a viable way to make use of the additional training data. (Also see comment about number of objects below).

Weaknesses

The paper is harder to read than needed. I appreciate there is a lot of stuff going, and I believe I was able to get most of it after a few iterations, so the lack of clarity is not extreme.

It is not clear how differing numbers of objects are handled. It seems that the number of objects might simply be provided (K=3 in the dataset)? But if the number of objects is known, then the comparison to other work that infers it might not be fair. K is in the pseudo code, but does not seam inferred. Should K be an input.

A few more details on the LSTM would help.

The paper could use some polishing. Figure 1, which is informative and key, could be tidied up. Also, I am guessing that the "Objects" box is the LSTM. The English could be improved in places, and there are a number of grammar errors (e.g., the first two uses of "pixel" in 2.3.1 should be "pixels", and 'This last two terms mean' on page 5).

The authors do not say whether they will release their code.

Comments

A clear limitation of this work is that the data is synthetic and very simple (although perhaps more complex than other work in this space). The authors acknowledge this in their discussion. While this might be the standard for this sub-area, real data from a robot or car should be relatively easy to get. Easy block-world-like real data might be better for pushing the work than adding more texture and diverse lighting in synthetic data.

Using just two time points is both a strength and a weakness. With just two, there is likely to be a lot of ambiguity between translation and rotation, especially if you generalize to more than one pose parameter. Looking at the effect of larger training sequences would be interesting.

While the part of the system that is deployed at testing is a simple network, there are a number of hand constructed components (e.g., the warping function) that make use of what we know about cameras, it would be more interesting to see those learned.


**Summary Of The Paper:**

Inspired by ideas about how humans learn about objects, the authors detail a system to train a neural network to perceive generic objects using image triplets where objects move and the viewer also can move.  The viewer's motion is provided as an input. Object perception by parts of the network operating on the first two time points is rewarded by predicting what is seen at the third time point (the training signal). Having been trained, the object perception part of the system, which is a relatively basic neural network, can segment them from a single image.  This new setup requires different data that what has been used in this space, and the authors contribute a synthetic dataset as well.


**Summary Of The Review:**

This is a good first step in this direction, and should inspire follow up work. The technical innovation is sensible. The results are good.

---

> ### Author Response · Authors · 2021-11-23
> **Thank you for appreciating the value of our work**
>
>
> Thank you for the positive review and for your insightful remarks. We really appreciate your understanding of the importance and our motivation for this work. We have tried to address the writing and clarity issues in the updated manuscript writeup.
>
> “It is not clear how differing numbers of objects are handled. It seems that the number of objects might simply be provided (K=3 in the dataset)? But if the number of objects is known, then the comparison to other work that infers it might not be fair. K is in the pseudo code, but does not seam inferred. Should K be an input. “ -- Yes, we set the value of K explicitly as the maximal possible number of objects we use in the scene. When there are fewer objects visible, the network outputs a few empty masks. In this sense, the network roughly “infers” K. We plan to test a setting where K is much bigger than the maximal possible number of objects later.
>
> “A few more details on the LSTM would help.” -- It is a generic LSTM. The initial state is set to zero. At each step, the vector output of the encoder is repeatedly fed as input, and the output at each step is treated as an intermediate code for each object, which further goes through a fully-connected network to infer location, pose and object representational code.
>
> Suggestions for the figures: thank you! Due to the time limit we have not incorporated your suggestions yet but will certainly include them in the final version.
>
> “The authors do not say whether they will release their code. “ -- Yes, we will be releasing our code upon publication.
>
> "A clear limitation of this work is that the data is synthetic and very simple...". Thanks for the suggestion. We have partially trained our networks on a much more challenging dataset of simulated car driving in a virtual town. After the rebuttal period, we plan to train on a higher-resolution version of it and in multiple towns under various weather conditions, and add the result to the final version.
>
> "Looking at the effect of larger training sequences would be interesting. " This is a good suggestion! We will consider this in a follow-up work. We take 2 frames as the simplest case to demonstrate the principle. And we found it actually learns to segment part of the border of cars correctly even when cars change speed and directions frequently in the Traffic dataset.
>
> "there are a number of hand constructed components ... it would be more interesting to see those learned. " We totally agree! We would like to test this idea in a follow-up work. Our goal is to eventually remove human supervision and hard-coded rules as much as possible to demonstrate how ingredients of symbolic representations can be learned from sensory data without supervision.

---

> > ### Comment · Reviewer_uocA · 2021-11-29
> > **This will be a nice paper, but not for ICLR 2022**
> >
> > As per my review, this is work heading in a nice direction. While I was the most positive reviewer, I was also worried about the somewhat unfinished state of the paper, and addressing some of the concerns of others requires substantive extra careful validation. I am impressed with the authors' effort to address all the comments. However, given the scale of the changes, this needs to go through another review process. So, while I am still positive about the work, I agree with the sentiment that this paper is not ready for ICLR 2022.

---

### Official Review · Reviewer_8qC3 · 2021-11-02

**Correctness:** 3
**Technical Novelty And Significance:** 3
**Empirical Novelty And Significance:** 2
**Recommendation:** 5
**Confidence:** 4

**Main Review:**

Overall, I found the paper quite interesting. I think the optical flow based warping to predict some subset of the pixels in the next timestep is in itself an important contribution and this paper would be a good addition to the emerging literature on unsupervised object-centric models.

I think the main concern with the paper is limited experimental evaluation. The model is evaluated only on a single, rather simple dataset (that was generated by the authors). I know these models cannot usually handle complicated datasets so it's fine to have a simple dataset but I'd have liked to see the model evaluated on some of the datasets that other competing models were evaluated on. Some of those datasets might not have camera information etc. but I'm sure there are other datasets that have the necessary information (e.g., see [1], [2] for some potential datasets).

Also, the authors mention that their technique is the first to infer 3D position of objects while segmenting images. If I'm not mistaken, [1] also does both and would be a great model to compare against.

Other notes

- Std deviations in table 1 are too large. Is this a typo? If not, it looks like all models are doing equally well.
- I don't have any specific recommendations here but I found the model description a bit hard to follow. It might be a good idea to do another pass and see if it can be organized better. For example, the fact that warping and imagination are combined to get the final image can be mentioned earlier so the reader knows where/how the imagination network is used.
- There are many inline equations; these make it difficult to parse the text visually. It would be nice to take these out of the text and split the long section 2.3.1 to subsections and mark these.
- Figure 1 is great but again hard to parse. Perhaps adding variable names (i.e., x, p, z etc.) to the figure might make it easier to understand what goes in and out of each network.

[1] Henderson, Paul, and Christoph H. Lampert. 2020. “Unsupervised Object-Centric Video Generation and Decomposition in 3D.” arXiv [cs.CV]. arXiv. http://arxiv.org/abs/2007.06705.
[2] Kabra et al., 2021. "SIMONe: View-Invariant, Temporally-Abstracted Object Representations via Unsupervised Video Decomposition", https://arxiv.org/abs/2106.03849

**Summary Of The Paper:**

This paper presents an unsupervised object-centric scene representation technique that can decompose a scene into multiple objects (and segment the scene) and infer their 3D locations and pose. The overall setup is very similar to earlier models like MONet but this model works on sequences of images, more precisely on 3 consecutive images. It uses the first two images to infer the 3D position and pose of objects and combining this with known camera motion tries to predict the last (third) image. The main contribution here is an optical flow based method to warp the image at time t using the predicted object location/pose/depth to predict (some of) the pixels in image at time t+1.

In more detail, the object extraction network outputs the location and pose of each object. And a separate depth perception network outputs the depth for each pixel in the image. The location and pose of objects are used to estimate the velocity of each object (e.g., by subtracting the position at t-1 from position at t. note this requires matching each object at time t-1 to object in time t, which they do using a soft-matching approach). These along with the depth information are then used to warp the image at t to predict pixels in image at time t+1. This is possible only for a subset of the pixels so for the rest, they use a separate "imagination" network that takes in object information and predicts the color/depth and object masks at t+1. The predictions from warping and imagination network are then combined to form the final predicted color and depth images.

To train the model, they require images and camera motion, and use a combination of losses: reconstruction loss on predicted and ground truth image, self-supervised losses on object location, pose, and depth.



**Summary Of The Review:**

Overall, I think the paper is quite interesting and would be of interest to the community. However, the empirical evaluation is very limited and this makes it difficult to evaluate the full merit of the proposed approach.

---

> ### Author Response · Authors · 2021-11-23
> **Thanks for your great suggestion. We have tested on new dataset now.**
>
> Thank you for the helpful feedback, we have fixed the writing and the clarity issues raised, and made the manuscript more structured.
> “Overall, I found the paper quite interesting. I think the optical flow based warping to predict some subset of the pixels in the next timestep is in itself an important contribution and this paper would be a good addition to the emerging literature on unsupervised object-centric models.”
> We are very grateful for acknowledging the importance of the works
> Below, we address your concerns in details.
>
> “I think the main concern with the paper is limited experimental evaluation. ... I know these models cannot usually handle complicated datasets so it's fine to have a simple dataset but I'd have liked to see the model evaluated on some of the datasets that other competing models were evaluated on. Some of those datasets might not have camera information etc. but I'm sure there are other datasets that have the necessary information (e.g., see [1], [2] for some potential datasets). “
> -- Thanks for your suggestion, we have now tested our model on the traffic dataset of Henderson & Lampert paper. And indeed, we are sorry for missing the reference to their work (O3V). We have removed the statement that we are the first in this domain, although with major distinctions from O3V (1. ours can generalize to single images while O3V’s architecture builds in alternating temporal and spatial convolution, which suggests it may require sequences of images at test time. 2., our model learns from prediction error while O3V learns from reconstruction loss in the VAE formulation. 3., we do not aim for image synthesis, although the imagination network can fill occluded parts)
> We have now evaluated our model on an extended version of the Traffic dataset used in O3V (in their paper, cars only move straight on one street. We now allow the cars to drive through the whole town). Although the training has not fully completed due to the time limit (we will update with more results if accepted), we provide a qualitative comparison of the predicted images and the segmentation or cars in the last figure of Appendix. Cars can be fully segmented and separate from other cars, although at this stage of training, we found the segmentation mask of our OPPLE network for the cars often extends two segments forward and backward onto the road surface. We reason this is because at the low resolution of the dataset (72x96), the road always has a homogenous color. Thus, mathematically, extending the mask of the car in this way does not increase prediction error (the cars don’t turn in the dataset). In the real world, even though we may occasionally see a cup in front of a white wall, we also often see it against other textured backgrounds such as dining tables. Therefore, we think this issue of homogeneous background on some sides of the objects in all data points is specific to this dataset.
>
> “Std deviations in table 1 are too large. Is this a typo? If not, it looks like all models are doing equally well.”
> -- We are sorry for the confusion. The original standard deviation we reported was calculated across different images from the same trained model. Obviously, because each model fails for some images or approaches perfect for other images, the standard deviation must be large. Recognizing that this number is rarely reported and causes confusion, we now only report the mean IoU as common practice in computer vision. After the rebuttal period, we will train the models multiple times and add the standard deviation across training results (some models such as a bigger version of slot attention take 5 days to train). Additionally, as included in our new reply to Reviwer iu2Y, the difference in ARIs and IOUs are highly statistically significant across models.
>
> “It might be a good idea to do another pass and see if it can be organized better. For example, the fact that warping and imagination are combined to get the final image can be mentioned earlier so the reader knows where/how the imagination network is used.”
> --Thanks for the suggestion. We have now moved part of the text earlier to section 2.2 (overall principle). There, we provide a description of how warping and imagination can work together to predict the next image. We have also moved many details into the appendix to help readers get the gist.
>
> “There are many inline equations; these make it difficult to parse the text visually. It would be nice to take these out of the text and split the long section 2.3.1 to subsections and mark these.”
> -- Thank you and we have taken your suggestion now to break the section into subsections and moved some equations to appendix.
>
> “Figure 1 is great but again hard to parse. Perhaps adding variable names (i.e., x, p, z etc.) to the figure might make it easier to understand what goes in and out of each network.”
> --Thanks for the suggestion! Due to the time limit, we will make this change at the final stage.

---

> > ### Comment · Reviewer_8qC3 · 2021-11-27
> > **Keeping my score**
> >
> > I'd like to thank the authors for their detailed feedback to my review. As I said in my original review, I think the paper is quite interesting and would be a good contribution to the field but unfortunately, even with the revisions to the paper, I don't think the evaluations make a strong case for the proposed technique so I'd like to keep my score as is. I'd very much encourage the authors to extend the evaluation (to other datasets and other models) and re-submit in the future if it doesn't make it through this time.

---

> > > ### Author Response · Authors · 2021-11-29
> > > **Thank you!**
> > >
> > > Dear reviewer,
> > > Thank you very much for the kind feedback. We will indeed add evaluation against the GQN dataset (currently in progress) in the update.

---

### Official Review · Reviewer_iu2Y · 2021-11-03

**Correctness:** 3
**Technical Novelty And Significance:** 2
**Empirical Novelty And Significance:** 2
**Recommendation:** 5
**Confidence:** 4

**Main Review:**

Overall, this paper is messily written, and proposes something that only works marginally better than prior methods, on a synthetic toy dataset. The performance in Table 1 illustrates this: the standard deviation of the segmentation IOU (0.34) is about even with the average IOU (0.35)! Looking at the qualitative results in Figure 2, it seems the model often misses objects completely, and produces segmentations that are fractional and have holes. I appreciate that the baselines are doing badly here also, but slot attention had very similar ARI-fg (0.42+-0.35 vs 0.46+-0.39). Statistically maybe these are equivalent. Is it possible at least to show that this method does better on the datasets where those previous papers managed to work (like CLEVR and those deepmind shapes datasets)? In general this area of work seems stuck in a setup where all methods work well on different toy data, but no methods work on real images or videos.

It seems like the pose estimation is not working at all, judging by Figure 3-E. The discussion mentions this too: "object spatial location is inferred more easily than object pose (which we have not fully investigated), thus the predicted warping relies more on object translation than rotation". Maybe the object pose estimation can be removed from the paper entirely, to make things simpler.

"our framework can easily generalize to 3D pose" I am sure the formulation can easily be extended to capture this, but I would prefer that the wording here be a bit more careful, to not suggest that the model is expected to work when pitch and roll are unknowns as well. (If you do expect it to work, please try it out and add the results to the paper.)

"At time t, given the location of the camera o(t) ∈ R3 and its facing direction α(t)" Does this mean the camera intrinsics and extrinsics are assumed known? It would be great to say this directly. The discussion section supports this interpretation, saying "additional information required is that of observer’s self-motion, which is available both in the brain as efference copy and easily accessible in vehicles". I do not really buy the argument about the brain, or easy accessibility in vehicles. Self-driving vehicles usually register themselves to a known map; inferring pose from odometry alone causes drift.

The LSTM reading objects from the full-frame encoding sounds like a very weak part of the model. Why not, for example, use a standard object detector, like MaskRCNN? I know you want to be self-supervised, but then why not self-supervise a well-known architecture that is proven to work, instead of inventing a new one?

"The location prediction is restricted to the range of possible value within the virtual environment by logistic function and linear scaling" I don't know what this means.

The imagination network is clumsily introduced. The first instance of it is already "the imagination network", and the motivation for it is only written in the caption of Figure1. I found a helpful description later on in page5. These things should be re-arranged.

The description of the unprojection of a 2D coordinate into 3D space looks odd to me. Where is this coming from? Given that i,j are coordinates, what do |i| and |j| represent? Normally we start from x=fX/Z, and then invert this to X=Zx/f, where x is 2D and X is 3D, and f is the focal length. I am also not sure how the first term and second term here are able to multiply, since the first term is a 2-tuple (i,j) and the third term is a 3-tuple (i,d,j).

I also got lost in the angular velocity equation. It seems the sum is over all \gamma1, all \gamma2, and all differences of the two within 2\pi of \omega. This is too many things to sum over! It seems like you won't end up with a valid probability distribution. I am probably misreading the notation here. In any case, why is this probability distribution useful? The object-based imagination network seems to require a scalar here for the rotational speed. So why not take the expectation of the first pose distribution, expectation of the second, and then take a difference?

It is interesting that you do not use depth supervision, but for toy settings like this I think it is OK to assume depth is known, and focus on other hard parts, like segmentation and tracking.

I need some help understanding L_{spatial}. The second term is described as a contrastive loss, but it's a difference between known positions and random positions. Why is this a good idea, and why is it contrastive? Normally a contrastive loss compares two estimates and pushes them apart (rather than pulling every estimate to random).

I was surprised that the evaluation talked about a model called "OPPLE" ("Only OPPLE shows a bi-modal distribution."). Apparently this is the name of the proposed model, and the place to learn this is the caption of Figure 1! Please do not put critical information exclusively in figure captions.


Typos:
- probability mess -> probability mass
- generated dataset -> generated a dataset
- states-of-art -> state-of-the-art
- network appear to -> network appears to
- intersectin (in fig3) -> intersection
- Figure 1 is never referred to in the text

**Summary Of The Paper:**

This paper presents a method to learn how to parse 3-frame videos into object-centric representations, which include segmentation masks and 3D positions and yaws for those objects frame-by-frame, as well as an image representation of the background, and an overall depth map. This is accomplished with a depth network, an object network with an LSTM at the bottleneck to iteratively pick out objects and their positions and yaws, and a decoder to provide segmentations, a warping/re-compositing operation that pastes the inferred objects at their estimated positions for the NEXT frame (i.e., with a constant-velocity assumption), and finally an "imagination" network which refines the estimated image. The model learns with a combination of 4 losses, which include image reprojection/prediction, depth consistency, a spatial term that includes consistency and randomness (though I have complaints about this), and finally a penalty term that discourages object probabilities from being zero. The paper also introduces a new synthetic dataset where prior methods do badly, and the proposed method does slightly better. The learned depth maps look good, but this is perhaps expected because camera poses are known.


**Summary Of The Review:**

I think the paper is not quite ready for publication. The method does not work particularly well, a part of the method (object pose estimation) seems to be not working at all, and the evaluation is only on a new toy dataset and does not include evaluation on established datasets. Also the text contains too much notation, and has some parts mixed up (with terms being used before they are introduced), but I think this can be fixed easily.

---

> ### Author Response · Authors · 2021-11-23
> **Thanks for your feedback. We have addressed your comments**
>
> Thank you for the meticulous and insightful review.
> Below are detailed replies
>
> standard deviation of IoU and ARI: The original standard deviation we reported was calculated across different images from the same trained model. We removed it as it is not commonly reported in computer vision literature and causes confusion.
>
> Missing objects and holes in mask
> --  This is due to the loss function of predicting future images: wrongfully predicting the motion of a small distant object introduces prediction error on only a few pixels which makes it difficult to learn for these objects. One remedy is to include sequences of images in which the camera gradually approaches an object from very far. Such observation is common in daily life as we approach objects. This again illustrates the importance of introducing datasets resembling what human eyes see. The reason for the segmentation to sometimes leave a hole is because for some objects of homogeneous color, when the motion is small, even treating the central part of the image as the background does not change the appearance of the predicted image. A wider range of object motion distance may resolve this.
>
>
> Other datasets:
> -- We now show qualitative performance on Traffic dataset and more details will be in final version.
>
>
> “In general this area of work seems stuck in a setup ...”
> -- We understand your frustration. We think the difficulty lies in the pursuit of unsupervised learning and the fundamental difficulty of developing symbol-like representation grounded in sensory data.
>
>
> We have now tuned down the wording about 3D pose to “for simplicity, we do not consider pitch and roll here and leave it for future work to extend to 3D pose”.
>
> Camera extrinsic-- We do not expect the ground truth of the extrinsic matrix to be known. We want to emphasize that the model does not require localizing the camera in the world. Only the intrinsics and the information of egomotion are needed.
>
> We have removed statement about vehicles. Example reference of efference copy includes Sommer, M. A., & Wurtz, R. H. (2008) Visual perception and corollary discharge. Also the vestibular sensors inside the ear detect head rotation.
>
> LSTM vs MaskRCNN --  LSTM is just one implementation choice we made. We are interested in demonstrating how the ability to “propose” object masks can be learned instead of being based on pre-specified region proposal algorithms. Though we would agree with your sentiment if the goal were for engineering application.
>
> “The location prediction is restricted to the range of possible values within the virtual environment by logistic function and linear scaling’  -> We have changed it to “The location prediction is restricted to the visible range of the camera with an upper limit of distance”. We use x = sin( fov/2*1.2 * (sigmoid(a)*2-1) ) * d_max * sigmoid(b), y = cos( fov/2*1.2 * (sigmoid(a)*2-1) ) * d_max * sigmoid(b), where a and b are two unbounded scalars output by a fully connected network on top of the LSTM. d_max is the range of possible distance, and the term within sin and cos restrict bearing angle in 1.2 x fov.
>
> “unprojection of a 2D coordinate into 3D space...“ -- |i|, |j| are the distance of a pixel to the vertical and horizontal line crossing the center of the image. In your notation, we assume you are using the definition that x,y are axes in the image plane and z axis is perpendicular to it. In this case, i, j, d in the equation correspond to x, y, f in your notation.
> Note that X,Y,Z are all unknown, except that the distance D of the pixel (in 3D) to the lens is inferred by the depth perception network. Because X^2+Y^2+Z^2 = D^2 , and because x/X = y/Y = f/Z, they should all equal to sqrt(x^2+y^2+f^2) / sqrt(Z^2+Y^2+Z^2) = sqrt(x^2+y^2+f^2) / D. We can thus obtain X/D = x / sqrt(x^2+y^2+f^2), Y/D = y / sqrt(x^2+y^2+f^2), and Z/D = f / sqrt(x^2+y^2+f^2). Then, multiplying D on both sides of these equation obtains X,Y,Z, as \hat{m(i,j)} in our equation.
>
>
> “I also got lost in the angular velocity equation. … ”
> -- The two sums are performed in separate steps. At each step, the resulting probability distribution is first normalized before the next step. Therefore, all distributions are guaranteed to sum to 1. To put it in words, the first sum is to estimate the distribution of rotational speed, marginalizing over all possible combinations of the first pose and the second pose. The second sum is to further marginalize over the probability distribution of the second pose and the distribution of the rotation to reach the probability of the third pose. The approach we take is fully Bayesian, thus maintaining uncertainty about each variable towards the end.
>
>
> X_rand is the estimated location of an object from a random image sample (within the same mini-batch). We have corrected the contrastive loss term (we missed a thresholding in the last version).
> We are indeed implementing what you suggested: pushing apart two estimates from two irrelevant images

---

> > ### Comment · Reviewer_iu2Y · 2021-11-29
> > **Thank you for the reply**
> >
> > I very much appreciate the wording changes and the clarifications, but based on the collection of comments and other reviews, I do not feel persuaded to change my score.
> >
> > I think the standard deviations in your tables does not "cause confusion". It reveals, as 8qC3 also mentioned, that perhaps all models are doing equally well.
> >
> > Also I appreciate that Z7dM and 8qC3 pointed out evaluation issues (as I did). It is necessary to at least evaluate on the same datasets as the baselines used, and not exclusively rely on new data, because maybe the comparison is favorable here but unfavorable in previously-studied setups. The rebuttal says the Traffic dataset has been added, but in the paper this looks purely qualitative, while I think the evaluation focus of the paper is more on accurate segmentation (as shown in Table 1).
> >
> > "We do not expect the ground truth of the extrinsic matrix to be known. We want to emphasize that the model does not require localizing the camera in the world. Only the intrinsics and the information of egomotion are needed."
> > OK, I guess you are defining "extrinsics" to be camera poses relative to some "world" coordinates, and you are saying that you "only" require camera poses relative to the first frame. To me, this doesn't make a difference. Perfect knowledge of camera poses is very difficult to acquire in real data with a moving camera, and the choice of coordinate system is irrelevant.

---

> > > ### Author Response · Authors · 2021-11-29
> > > **statistical significance**
> > >
> > > Thank you very much for replying to our rebuttal. We really appreciate it.
> > >
> > > To address the concern about the standard deviation, because all methods were evaluated on the same set of testing images, we are able to perform paired t-test on the object-wise IoU underlying the original standard deviation we reported in the first version.
> > > Compared to MONet-128, our model is significantly better (T=36.5, p<6e-271, paired two-tailed t-test). Compared to GENESIS, our model is also significantly better (T=89.2, p=0, paired two-tailed t-test). You can also qualitatively observe the difference in the distribution of IoU in Figure 3A-D. We won't deny that slot-attention-128 now outperforms our model in IoU(T=45.2, p=0) but not in ARI, but as we mentioned, the proposed model is more explicitly 3D-aware.
> > >
> > > In terms of ARI, paired two-tailed t-test shows our model outperforms all other models: T=19.5, p<9e-81 against MONet-128, T=12.4, p<1e-34 against slot-attention-128, and T=42.5, p=0 against GENESIS.
> > >
> > > We understand your concern about obtaining perfect knowledge of camera motion, and we do also agree that in practice perfect motion parameters are impossible to obtain. We will provide new results trained with noisy camera motion parameters in the update to address this concern. Current work is a demonstration of principle, and ground truth camera motion parameters are used in several other important works, including O3V and GQN.
> > >
> > > We are in the process of evaluating against GQN dataset and will include in the update.
> > >
> > > We appreciate your spending time providing feedback to our paper regardless of your final decision.

---

### Official Review · Reviewer_Z7dM · 2021-11-03

**Correctness:** 4
**Technical Novelty And Significance:** 2
**Empirical Novelty And Significance:** 3
**Recommendation:** 3
**Confidence:** 3

**Details Of Ethics Concerns:**

No concerns.

**Main Review:**

# Strength

1. The paper tackles the difficult problem of learning to segment objects from an image using no supervision during training.
2. The problem setting and motivation for this task are explained clearly. A detailed description of the method, along with a pseudocode of the learning algorithm is provided in the paper.
3. The paper introduces a new synthetic dataset of images taken from scenes with multiple objects with varying shapes and textures (11 and 15, respectively).
4. Figures 3A-D are very helpful explaining the quantitative performance of the method in relation to the baselines. Figures 3E-G are also helpful showing the failure modes of the proposed method.

# Weakness
1. I am not fully convinced that the comparison to the baselines is entirely fair. If I understand correctly, the rest of the methods were trained on single images without having access to previous and next frames. While I appreciate the method’s usage of consecutive frames as part of the supervision, I think this should be stated clearly when comparing with the baselines -- to avoid any overclaims.

    SynSin [1], for example, also predicts the next frame’s RGB and depth images without using additional supervision. Similar to the proposed method in this paper, SynSin synthesizes future images given their camera poses by warping the current frame using differentiable rendering. Combining an object-centric approach (like slot-attention) with such a method that performs future image prediction would make a fairer comparison, in my opinion.

    [1] Wiles, O., Gkioxari, G., Szeliski, R. and Johnson, J., 2020. Synsin: End-to-end view synthesis from a single image. In Proceedings of the IEEE/CVF Conference on Computer Vision and Pattern Recognition (pp. 7467-7477).

2. I think there are some key references missing in the paper too. For instance, a paper from last year also learns an object-centric representation in an unsupervised fashion to decompose the objects in a scene while estimating their poses in 3D [2]. How does [2] compare to the method presented in this paper? What are the main differences between them?

    [2] Henderson, P. and Lampert, C.H., 2020. Unsupervised object-centric video generation and decomposition in 3D. Advances in Neural Information Processing Systems, 33.

3. The results are reported using only the dataset introduced in the paper. I suggest including results from datasets like the Objects Room or CLEVR, where the existing methods (MONet, Slot-attention, Genesis, [2]) have already reported results on.

4. There are no ablation studies reported in the paper. Some parts of the loss function seem redundant judging from their current descriptions. I think it would be very helpful either presenting additional results by ablating the loss function / part of the model, or detail in the text why the method needs each of those components.

# Recommended Decision

I think in its current form the paper is not ready to be published. I strongly encourage the authors to clarify the positioning of this paper in relation to the state-of-the-art (see my comment on weaknesses). I vote for [reject, not good enough] for now, but I would be happy to increase my rating once the authors clear up my concerns.

# Additional Feedback

1. It should be \hat(m) instead of \hat(t) in the L_spatial loss’ third term.

2. What is x_rand? If they are randomly sampled point positions then what is their underlying distribution?

3. Have you tested the method on scenes with more than 3 objects? Slot attention, for instance, is able to segment up to 9 objects. Since the paper introduces a new dataset, I would’ve hoped to see a more challenging benchmark with a wider variety of objects and number of instances.

4. Also, the dataset assumes that the camera moves with constant pose change. Have you tested the motion prediction (position, orientation and their time-derivatives) in the presence of different camera velocities? If not, are there any limitations of the method that prevents it?

5. I suggest adding the equivalents of Figure 3G for each baseline in the appendix.

6. What is the number of bins for the yaw angle prediction, b? Have you tried using a continuous representation for the rotation?

7. What does the predicted image (I’) for the next time-step look like? I suggest including more qualitative results to the paper for evaluating the warping function.

8. I think it is a good idea to introduce a more diverse benchmarking dataset for learning object-centric representations. However, I think the dataset proposed in the paper should be further expanded by including more daily life objects, instead of just geometric primitives like spheres, prisms and cones. I suggest taking a look at datasets like YCB or ShapeNet to include more realistic objects to your dataset.


**Summary Of The Paper:**

This paper studies the problem of predicting the segmentations and poses (position + yaw orientation) of multiple objects, given the image of a scene. The paper introduces a method that is trained without supervision for the segmentations, similar to several other recent object-centric models. In contrast to these existing models, the method proposed in this paper additionally estimates the 3D location of each object by predicting a depth map and classifies the yaw angle by representing the pose domain as equally-spaced bins. To do so, during training the method operates on a short clip of the scene recorded by a moving camera, and uses self-supervision by predicting the scene’s image at the next time step. At test time, the model is able to infer a representation of each object in the scene and segment them given a single image of the scene.

**Summary Of The Review:**

All in all, I think the paper attempts to tackle a very challenging problem. The method looks sound and the results might be interesting to the community in learning object-centric representations. However, there are some major concerns I have, mainly about the position and novelty of this paper with respect to a paper from last year, and the lack of results from datasets the state-of-the-art methods report on.

---

> ### Author Response · Authors · 2021-11-23
> **Thank you for your constructive review. We have clarified on the comparisons and added results with another dataset.**
>
> We appreciate the constructive review and your understanding of the difficulty of the problem setting. As pointed out by Reviewer uocA, even though the general idea inspiring our work has been around in psychology, such a principle (unsupervised learning object-centric representation with 3D awareness by prediction) has not been demonstrated on deep networks before (we will discuss the difference from [2] later).
>
> 1. The suggestion of integrating a slot-attention mechanism to [1] is certainly an interesting direction. However, such an architecture will be a standalone work worth publication per se, instead of serving as a simple baseline (as such a work has not been proposed elsewhere). The focus of [1] is a new 3D-aware view synthesis method. We hope to clarify that view synthesis is not the goal of this paper, but a tool for learning object-centric representation (by providing teaching signals). Although we demonstrated the principle of learning object-centric representation by predictive learning, the exact approach for rendering future images is not restricted to warping the RGB colors directly (as we did), as long as it utilizes 3D geometry. The rendering approach by Synsin can be one alternative (advanced) approach. We think that demonstrating the principle with a simpler rendering approach as we did can help readers focus on our major contribution. We now discuss the combination of the two as a future direction in Section 4.
>
> 2. Thank you for pointing out the missing reference for O3V [2]. We now add a comment in Section 4 on one major difference: O3V’s architecture alternates between spatial and temporal convolution. Thus it requires sequences of images at test time. All the three networks in our method are built to process each image independently. Even though the learning process requires sequences of images, the networks maintain the ability to segment and localize objects for a single image at test time. Another seemingly subtle difference is that our model learns from prediction while O3V learns from reconstruction. Reconstruct all pixels is demanding for the representational bottleneck, while our model can utilize the existing frame to predict part of the next frame so that the networks can concentrate on learning the common features defining object boundaries and spatial dependency in commonly observed scenes (imagination).
>
> 3. We have added our comparisons with the Traffic dataset. Qualitatively it can successfully segment cars with limitations (cf. general reply above). With more training, we expect further improvement and will report details in the final version.
>
> 4. We will add ablation studies in the final version.
>
> To additional feedbacks:
>
> 2. X_rand is the inferred location of the same slot output by the LSTM, uniformly sampled from all images. In practice, this is implemented by shuffling the inferred locations within the same mini-batch. We have made it clearer.
>
> 3. As a demonstration of principle, we have not tested on scenes with more objects. However, the architecture can in principle incorporate more than 3 objects by running the LSTM for more than 3 steps. We appreciate this suggestion but would like to point out that many other models (including O3V) have only tested on scenes with 4 objects and 4 cars at most. When the camera is close to objects, in practice it is difficult to see many moving objects.
>
> 4. This is a good suggestion. We now test our model on an extended version of the traffic dataset from [2], where the cars frequently accelerate and decelerate due to traffic control and distance to the car in front. We now also allow cars to turn (which was disabled in [2]). Both these introduce changing speed and angular velocity. Because our model makes predictions based on only 2 frames, it is impossible to estimate acceleration and some mis-prediction must occur. However, because motion is mostly smooth on the street, it appears that the model still manages to learn segmentation and depth (we have not fully investigated localization due to the limit of time). Of course, one natural future extension is to incorporate more frames and learn more complex dynamics for prediction.
>
> 6. The number of bins is 120, which means each bin covers 3 degrees. We chose this approach because using a continuous representation has difficulty in gradient passing - 0 and 360 degrees have no difference but the variable representing pose angle will have a big jump here. One possible solution is to separately parametrize cosine and sine of the pose while restricting their squares to sum to one. We leave this for future work.
>
> 7. Please refer to the new Appendix A.6
>
> 8. Our dataset includes some more complex-shaped objects like the torus, monkey heads, and the bunnies. Most existing object-centric representation works have used only the simpler primitives in our dataset and ours is already an extension. By reusing many shapes from other datasets being tested so far, we strike a fair comparison.

---

> > ### Comment · Reviewer_Z7dM · 2021-11-29
> > **thanks for the responses**
> >
> > Thanks to the authors for their detailed responses to the reviews. I appreciate that the paper is in the process of having an extended evaluation and a better positioning with respect to the state-of-the-art.
> >
> > I think pointing out the differences between the proposed method and O3V [2] was important, thanks for including them in the related work section. However, given its high relevance, I believe it is necessary to add comparison results against O3V on the Traffic dataset in the next version of the paper. Other than that, I agree with the reviewers iu2Y and 8qC3 about their comments on other issues in evaluations.
> >
> > To summarize, after reading the rebuttal I think the paper is going in the right direction. However, it needs a bit more work to report the comparisons and findings more clearly. I believe the paper would make a much stronger case in its next version after addressing the concerns brought up in the reviews.

---

### Author Response · Authors · 2021-11-23
**Thank you for all the comments. New evaluation added.**

We thank all the reviewers for their insightful and meticulous feedback. We have restructured and re-written some parts of the manuscript that were pointed out as terse or unclear.

Several reviewers pointed out Henderson & Lampert 2020, a highly relevant paper. We are sorry for missing the reference. We have now partially trained our networks on a more challenging version of their Traffic dataset - video of cars running in a virtual town. Qualitative illustrations are provided at the end of the appendix. At the current stage of training, the networks can successfully parse cars from the background and allocate different cars to different slots. However, the learned mask often assigns some probability to part of the road surface as belonging to the car. We think this is due to the fact that the images are sampled at a low resolution (96x72 as in the original paper) and that the color of the road is close to homogenous. First, this makes it difficult to learn the accurate depth of the road surface. Second, because within the dataset, the front and back borders of the cars are always surrounded by a close-to-homogeneous gray, there is little feedback for the model to learn a correct boundary. Because this situation is very unnatural compared to the visual experience of animals and humans, we think this is an issue of the data which may be resolved by using an even more realistic and higher-resolution dataset. The fact that the network has no problem learning the top border of the car illustrates this point.
Note that the original dataset was simpler than the version we generated (in the original version, cars can only move on a single street while we have allowed the cars to drive through the whole town). We think this reflects the strength of our model: the more complex the scene, the better segmentation it may learn. Full details of the comparison will be added in the final version. We think the existing results have already demonstrated the unique strength of the model.
We would like to point out one distinct feature between our model and O3V (Henderson & Lampert). All three subnetworks of our model process each image independently during training. Therefore, they maintain the ability to segment objects and infer 3D structure on single images. O3V alternates temporal and spatial convolution in its architecture. Therefore, it is unclear how it performs on processing a single image after training. Because the O3V's decoder is forced to generate the background and foreground (due to the VAE framework), it is possible that a much larger version of the network is needed to learn from an entire town. Because our model does not require this (imagination is based on part of the observed image), it does not need to remember every details of the images and may learn a more general implicit knowledge of how to segment objects. As demonstrated in the Appendix, it has no problem learning from videos in an entire town.

Other comments are replied to in more detail individually. We highly appreciate your feedbacks which has helped make our paper stronger and clearer.

During the review period, we have trained a bigger version of slot attention model for a longer time, and its performance improved. Therefore we updated the results. Our model still performs the best on ARI, and stays second on IoU. However, we would like to emphasize that in addition to segmenting objects, our model can explicitly estimate 3D location of each object, which is not available in the slot-attention model.

---

### Decision · Program_Chairs · 2022-01-20

**Decision:**

Reject

**Comment:**

This paper tackles the difficult problem of learning to segment objects from an image using no supervision during training. The paper is clearly written and a new synthetic dataset is made available. Unfortunately, the reviewers raised a number of issues with the submission (missing citations and comparison to relevant related work / additional baselines  + ablation studies / missing empirical evaluation of the proposed method on standard dataset beyond the toy dataset proposed by the authors). The paper received 1 reject, 2 marginal rejects and 1 accept but even the positive reviewer agreed that these were limitations. The authors also conceded to these limitations and initiated experiments that are starting to address the reviewers' comments. At this time, the results of these experiments remain incomplete and hence most reviewers agree that the paper should go through another round of reviews before it is publishable. I thus recommend this paper be rejected in the hope that a subsequent revision will make it a much stronger contribution.